

# Statistical analysis of global ocean significant wave heights from satellite altimetry over the past two decades

Alice Laloue[1], Malek Ghantous[1], Yannice Faugère[1], Alice Dalphinet[2], Lotfi Aouf[2]

[1]Collecte Localisation Satellites, Ramonville-Saint-Agne, 31520, France
[2]Météo-France, Toulouse, 31000, France

*Correspondence to*: Alice Laloue (alaloue@groupcls.com)

**Abstract.** The analysis of global ocean surface waves and of long-term changes is important to climate research and to ocean and coastal applications. Indeed, waves contribute to flooding, coastal erosion, extreme sea level events and ocean circulation. They also play a role in air-sea and sea-ice interactions. However, this analysis requires accurate time series of
waves over several decades that have previously only been available from model reanalyses or from in situ observations. Altimetry provides a long series of observations of significant wave heights (SWHs) in the global ocean. The aim of this study is to analyse the climatology of wave heights and extreme wave heights derived from remote sensing in the global ocean and their long-term trends from 2002 to 2020 using different statistical approaches as the mean, the 95[th] percentile and the 100-year return level of SWH. The mean SWH and the 95[th] percentile of SWH were calculated for two seasons: January,
February and March (JFM), and July, August and September (JAS) and for each year. A trend was then estimated using linear regression for each cell in the overall grid. The 100-year return levels were determined by fitting a Generalised Pareto distribution to all exceedances over a high threshold. The trend in 100-year return level was estimated using the transformed-stationary approach proposed by Mentaschi et al. (2016), which, to our knowledge, was used for the first time to draw a global map based on altimetry. Predominantly large positive trends over 2002-2020 for both SWH and extreme SWH are
mostly found in the southern hemisphere, such as in the South Atlantic, the Southern Ocean and the southern Indian Ocean, which is consistent with previous studies. In the North Atlantic, SWH has increased above 45°N, corroborating what was concluded in the fifth IPCC Assessment Report, however SWH has also largely decreased below 45°N in JFM. As for 100-year return levels of SWH in the northern hemisphere, they have significantly increased in the North Atlantic and in the eastern tropical Pacific, where the cyclone tracks are located. Finally, in this study we find trends of SWH and 95[th] percentile
of SWH for JFM and JAS over 2002-2020 to be much higher than those indicated in the literature for the period 1985-2018.

## 1 Introduction

Increasing our understanding of global ocean surface waves, their variability and their long-term interannual changes is important to climate research and to ocean and coastal applications. As mentioned in the sixth IPCC Assessment Report, waves contribute to extreme sea level events (Mentaschi et al. 2017), flooding (Storlazzi et al., 2018) and coastal
erosion (Barnard et al., 2017). They modify the ocean circulation and mediate air-sea (Donelan et al., 1997) and sea-ice interactions (Thomas et al., 2019).

The analysis of long-term and interannual changes of ocean surface waves requires accurate time series of waves spanning several decades that have only been available so far in global model reanalyses or from in situ observations. Unfortunately, using observations from buoys can only provide local analyses and in situ wave observations are especially
lacking in the southern hemisphere. Altimeters offer global and high-quality measurements of significant wave heights (SWH) (Gommenginger et al. 2002). The growing satellite record of SWH now makes global and long-term analyses more accessible than ever before.

We use SWH observations from a multi-mission altimetric product over the period 2002-2020 to calculate global SWH and extreme SWH climatologies. Furthermore, trends in SWH and in extreme SWH are assessed. An identical analysis
was performed with ERA5 (Herbasch et al., 2023) and WAVERYS (Law-Chune et al, 2021) reanalyses to compare with the





literature. The multi-mission nature of our altimetric data, and their potential for bias, is then discussed in the context of long-term statistics.

## 2 Sea State datasets and methods

The level 4 (L4) altimetric time series of waves in the Copernicus Marine catalogue (product reference 1) covers 19 years (2002-2020). It is based on Copernicus Marine Service multi-year L3 datasets and merges along-track measurements from 7 different altimetric missions - Jason-1, Envisat, Jason-2, Cryosat-2, Saral/AltiKa and CFOSat - and from up to 4 missions at the same time. Calibrated and filtered along-track measurements are then projected onto a 2° grid. Daily statistics (mean, maximum) are finally estimated for each grid cell.

We use this time series to calculate mean and extreme SWH climatologies and to assess long-term trends over the period 2002-2020. Meanwhile, the annual anomaly for 2022 is calculated as the difference between the climatology and the near-real-time time series (product reference 2). The first part of our analysis is based on daily mean SWHs and $95^{th}$ percentile ($P_{95}$) daily maximum SWHs over the globe. The $95^{th}$ percentile is the value above which 5% of the values in the time series fall. Data are resampled in monthly mean and percentiles of SWH for each grid cell. The climatological mean SWH and $P_{95}$ are calculated for both January, February, March (JFM) and July, August, September (JAS) separately to take seasonal variability of waves into account.

Trends in daily mean SWH and in $P_{95}$ daily maximum SWH were determined for each grid cell and examined in more detail for specific regions with large and significant trends. Trends were assessed using linear regressions, applied separately on the two seasonal datasets (JFM and JAS) as in (Timmermans et al., 2020), the significance of the resulting slopes were then tested at the 5% level using a Wald Test with t-distribution of the test statistic.

The second part of the analysis is focused on determining 100-year return levels that are likely to be exceeded, on average, once every hundred years (Goda, 2000) using the non-seasonal transformed-stationary approach (Mentaschi et al., 2016) and on assessing their trends. The extreme value analysis (EVA) consists in modelling the SWH with a statistical distribution and in estimating return levels associated with long return periods and small probabilities of occurrence. The EVA allowed us to study 100-year SWH with only a 19-year long altimetric time series. All the values of SWH exceeding the $95^{th}$ percentile and separated by at least 72h were selected according to the peaks-over-threshold method. A Generalised Pareto Distribution (GPD) could then be fitted to the exceedances. The return levels associated with the 100-year return period were estimated from this GPD.

The EVA has a major disadvantage in that it usually requires the time series to be stationary. The transformed-stationary approach overcomes this issue by transforming the non-stationary altimetric time series into a stationary one. It first standardises the series, then makes the location and shape parameters of the GPD time dependent. The results are finally transformed back into a non-stationary extreme SWH distribution, enabling us to assess the trend in extreme SWH over the entire period. To our knowledge, while this method has already been applied to ERA5 reanalysis (Takbash et al., 2020), it hasn't been applied to altimetry at a global scale before, thus only results obtained using ERA5 can be compared with the literature.

Finally, the same study was conducted for SWH from the ERA5 (Herbasch et al., 2023) and WAVERYS reanalyses (Law-Chune et al, 2021), as they allow for comparison with the literature (Timmermans et al. 2020) and the L4 altimetric time series.





## 3 Results

Climatologies of SWH and of high SWH are shown in Figs. 1a and 2a respectively. Energetic conditions in the
northern hemisphere, driven by extratropical storms, occur predominantly in the midlatitudes, reaching up to 4.5-5.0 m on
average in the North Atlantic and 4.0-4.5 m in the North Pacific during the JFM seasonal average. This contrasts with the
seasonal average during JAS that reveals corresponding energetic conditions in the Indian Ocean and in the Southern Ocean
up to 5.0-5.5 m, along with other seasonal processes such as the Asian monsoon, manifesting in increased wave height in the
Arabian Sea and in the Bay of Bengal. The spatial structure of the $P_{95}$ of SWH is consistent and shares similar patterns with
those seen in Fig. 1, with greater magnitude. Indeed, the highest SWHs can locally reach up to 9.0-10.0 m in the North
Atlantic and up to 8.0-9.0 m in the North Pacific (both in JFM) and up to 8.0-10.0 m in the Indian Ocean in JAS. Other
energetic conditions associated with typhoons are also revealed in the Philippine Sea, leading to high SWH of up to 6.0 m in
JAS. Smaller regional scale seasonal phenomena are also evident despite the poor spatial resolution of altimeters, such as
waves of 4.0-5.0 m in the eastern Pacific driven by Thuantepecer events at the Chivela Pass in JFM.
Trends of SWH and of $P_{95}$ of SWH are displayed Figs. 1c and 2c respectively. Some of these trend patterns have
already been described in previous studies (Young and Ribal 2019; Shimura et al. 2016). In general, large and significant
trends appear to be concentrated in the southern hemisphere: in the Southern Ocean, in the sector south of Africa, in the
Indian Ocean south of Australia. Young and Ribal (2019) had already highlighted the existence of a broad region of positive
and significant trend in the 90th percentile of SWH across the Southern Ocean with altimetric data spanning 1985-2018.
Patterns associated with positive trends in SWH and in the $P_{95}$ of SWH south of Africa, south of Australia and in the South
Pacific seem to mostly coincide with this broad region, as well as the decreasing SWH in the Indian Ocean around 45°S.
However, in contrast with Young and Ribal (2019), our trend in $P_{95}$ of SWH in the North Atlantic is not as significant and
positive. Moreover, significant trends are found in wintertime in forms of complex spatial patterns of increasing and
decreasing wave heights in the North Atlantic and North Pacific. In agreement with Young and Ribal (2019), SWH in the
North Pacific shows a distinct negative trend that is especially true in our case during wintertime. The negative trend of
SWH in JFM in the western North Pacific also agrees with the decreasing winter wave heights in global climate models
(Tomoya Shimura et al. 2016). As in (Timmermans et al., 2020), significant positive trends are also found in the North
Atlantic and in the Gulf Stream region. Finally, the results depicted in Figs. 1 and 2 suggest that significant upper percentile
trends are changing orders of magnitude faster than trends of mean SWH.
Anomalies of SWH and of $P_{95}$ of SWH for 2022 are shown Figs. 1b and 2b respectively. The average interannual
variability of wintertime SWH is of the order of 0.13 m at tropical and subtropical latitudes and 0.30-0.40 m at mid-latitudes,
with regional excursions exceeding 0.40 m, while the interannual variability of extreme wave heights averages 0.33 m at
tropical and subtropical latitudes and 0.70-0.80 m at mid-latitudes, with regional excursions exceeding 0.85 m in summer in
the typhoon region and 0.90 m in the Southern Ocean. Despite this high interannual variability, some SWH anomalies for
2022 are found to exceed it and some do not appear to be stand-alone events, but rather part of longer-term changes in SWH
(Figs. 1c and 2c). Strong positive anomalies found in the North Atlantic and North Pacific around 60°N in winter and for
both SWH and $P_{95}$ of SWH mostly coincide with increasing SWH and $P_{95}$ of SWH. While the negative anomalies in the
North Atlantic and in the North Pacific may not exceed the interannual variability, they still partly coincide with
corresponding trends.







**Figure 1: SWH (a) climatology (2002-2020), (b) annual anomaly for 2022 and (c) annual trend (2002-2020), for both JFM (left column) and JAS (right column) from L4 altimetric time series of daily mean SWH (product reference 1). Areas with anomaly above 1.5 times the interannual variability are outlined in black. Areas with trend statistically significant at the 95% level are outlined in black.**





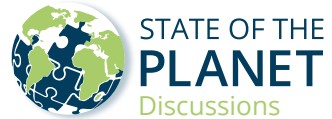

**Figure 2: 95th SWH percentile (a) climatology (2002-2020), (b) annual anomaly for 2022 and (c) annual trend (2002-2020) for both JFM (left column) and JAS (right column) from L4 altimetric time series (product reference 1). Areas with anomaly above 1.5 times the interannual variability are outlined in black. Areas with trend statistically significant at the 95% level are outlined in black.**

The most energetic conditions on the map of 100-year return levels can be found on a large scale in the North Atlantic and western North Pacific driven by extratropical storms and typhoons, on a smaller scale in the eastern tropical Pacific and Indian Ocean by hurricanes and tropical cyclones. As expected, the strongly positive trend patterns found in the southern hemisphere are consistent with those highlighted by the SWH and the $P_{95}$ of SWH. However, despite the very contrasting trends of SWH and of $P_{95}$ of SWH, the trends of 100-year return levels are largely positive in the North Atlantic. Certain regions also stand out with a very significant trend, in contrast to that observed in the $P_{95}$ of SWH, such as the western North Pacific, the Gulf of Mexico and the Caribbean Sea, which demonstrate strong negative trends contrary to what Takbash et al. (2020) found. On the other hand, as shown by Takbash et al., localised positive trends can also be found in the





hurricane regions in the tropical eastern Pacific and in the typhoon regions; these increases were not visible in the trends in SWH and P$_{95}$ of SWH either.

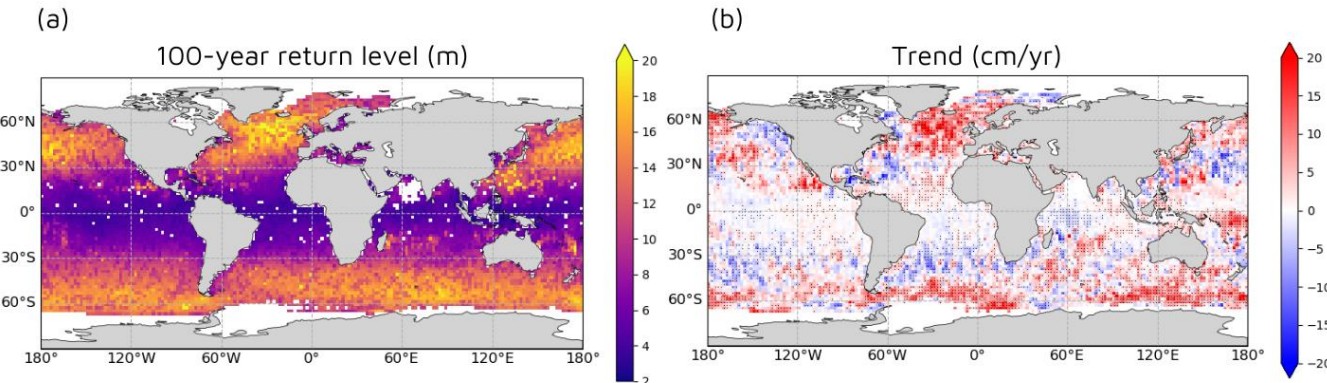

**Figure 3: Average 100-year return levels and their trends over 2005-2018 from L4 altimetric time series (product reference 1) using the non-seasonal transformed stationary approach. Areas with trend statistically significant at the 95% level are indicated by grey dots. White pixels correspond to grid cells that do not meet the requirements for calculating return levels, such as the minimum number of points selected with the Peaks-over-threshold method.**


For comparison, the same figures were produced using ERA5 and WAVERYS data. Spatial patterns are mostly consistent between all three datasets, however ERA5 and WAVERYS slightly underestimate the mean SWHs and their trends. As for extreme values, they overestimate the highest values and trends of P$_{95}$ of SWH, especially in the southern hemisphere, but they underestimate 100-year return levels compared with altimetry as noted by Takbash et al. (2020).


## 4 Discussion

In this study, we found large positive trends over 2002-2020 for both SWH and extreme SWH, mostly in the southern hemisphere, which are consistent with findings by Young and Ribal (2019). In the North Atlantic, SWH has increased above 45°N, corroborating what was concluded in the fifth IPCC Assessment Report (AR5), and in 2022, a large positive anomaly of SWH and extreme SWH was found in the same region. However, SWH has also largely decreased below 45°N in JFM, contrary to Young and Ribal's (2019) findings. As for 100-year return levels in the northern hemisphere, they have significantly increased in the North Atlantic and in the eastern North Pacific, where the cyclone tracks are located. Finally, we found trends of SWH and P$_{95}$ of SWH for JFM and JAS over 2002-2020 to be much higher than those indicated by Young and Ribal (2019), for the period 1985-2018.



The mean trends of SWH and P$_{95}$ of SWH were estimated for regions where the trend in the grid cells was predominantly statistically significant in the multi-mission product (see Fig. 3a), such as in the North Atlantic (boxes 1 and 2), in the South Atlantic Ocean and Southern Ocean (box 4), in the Southern Indian Ocean (box 5) and in the western North Pacific (box 3). In JFM, the SWH increases by 1.8 ±1.1 cm each year above 45°N and decreases by 2.1 ±0.76 cm each year below 45°N in the North Atlantic. In box 4, the SWH increases by 1.8 ±0.41 cm each year in JFM and 1.2 ±0.61 cm each year in JAS, and the P$_{95}$ of SWH increases by 3.5 ±1.9 cm each year in JFM. Finally, the P$_{95}$ of SWH increases by 3.1 ±1.7 cm per year in JFM in box 5.


Unfortunately, no uncertainty is provided for the SWH data from the multi-mission product, so only an uncertainty on the trend adjustment could be calculated. The major concern regarding the estimates of the trends of daily mean SWH and P$_{95}$ daily maximum SWH is the fact that the number of satellites combined in the multi-mission product has increased over time (Charles, 2021). This concern was previously addressed by Young and Ribal (2019) in relation to their own multi-




mission altimetric product. With more satellites, the number of along-track measurements available from which daily statistics are estimated and the number of days available increase. Consequently, daily statistics are more frequent and precise at the end of the period than at the beginning. For example, it is likely that more storms or extreme waves were sampled by the altimeters in the latter years of the period than in the former hence producing a spurious positive trend in SWH. Moreover, due to the polar altimeter orbits, the median number of annual observations also varies with latitude from a minimum at the equator to a maximum at latitudes above 60°N.

A series of tests were performed to evaluate the effect of the increasing number of satellites on the trends. A new L4 altimetric time series was created by combining only two satellites at a time to serve as a means of comparison for the L4 multi-mission product. This new product only extends to 2019, so the two products were compared over the period 2002-2019. The SWH trends that are statistically significant for both products are plotted in Fig. 3. The time series differ from each other starting from 2008 with the introduction of more satellites in the multi-mission product, whereupon the number of observations doubles (Figs. 3b, 3c, 3d). The mean SWH is not greatly affected by the number of satellites and the trends of mean SWH are almost identical. However, the $P_{95}$ daily maximum SWH is sensitive to the increase in the number of observations and the multi-mission product overestimates its trends compared with the two-satellite product. However, the sign of the trend doesn't change, and the spatial patterns of the trend are mostly consistent between the products. More importantly, trends in the two-satellite product are contained within the uncertainty of trends in the multi-mission product.

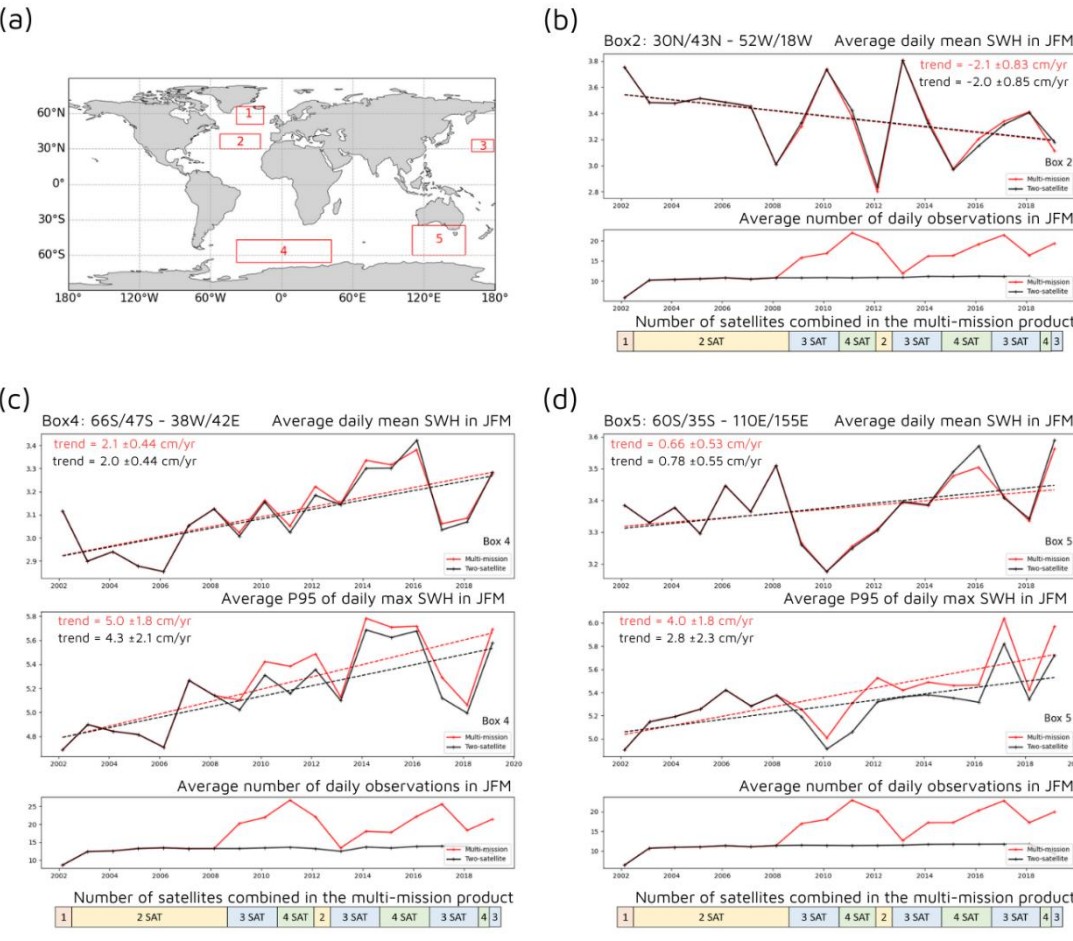





**Figure 4: Effects of the number of satellites on the long-term trends in L4 altimetric time series. (a) Boxes in which regional trends were computed. Box 1: 51° N-66° N, 28° W-15° W; box 2: 30° N-43° N, 52° W-18° W; box 3: 28° N-38° N, 160° E-180° E; box 4: 66° S-47° S, 38° W-42° E; box 5: 60° S–35° S, 110° E-155° E. (b), (c), (d) Time series of daily mean SWH, of $P_{95}$ daily maximum**
**SWH, and of the daily number of observations in JFM averaged on a yearly basis, associated with Boxes 2, 4 and 5 respectively. In red: the L4 multi-mission product (product reference 1), in black: L4 two-satellite product. Trends are represented by dashed lines when statistically significant for both products. Finally, the number of satellites combined in the multi-mission product is represented by coloured blocks as a function of time as in (Charles, 2021).**

There is a strong positive trend in the southern hemisphere which has also been observed in other studies and in reanalyses, however the altimeter observations have been calibrated and validated using in situ observations almost entirely
located in the northern hemisphere and near coastlines (Charles and Ollivier, 2021), potentially biasing the altimetry record. Although the trends should themselves be robust, caution should nevertheless be exercised in interpreting this result until such a time as more southern hemisphere and open-ocean in situ observations can be included in the calibration.

**5 Conclusion**

We have derived global ocean wave and extreme wave height climatologies and their trends for the period 2002-
2020 based on the mean, the 95$^{th}$ percentile and the 100-year return level of SWH from an L4 altimetric time series. To our knowledge, this is the first time that a global 100-year return level trend map has been drawn from an altimeter series using the transformed-stationary method. The climatologies and trends computed from satellite altimetry were very similar to ERA5 and WAVERYS.

Over the last two decades, predominantly large positive 2022 anomalies of SWH and significant 2002-2020 trends
are mostly found in the southern hemisphere. Large significant positive trends in mean SWH and $P_{95}$ of SWH are found in the South Atlantic, the Southern Ocean and the southern Indian Ocean (up to 1.2 ±0.61 cm/year for the SWH, up to 3.5 ±1.9 cm/year for the $P_{95}$ of SWH). According to the AR5, as winds are likely to strengthen in the southern hemisphere, this trend could be confirmed in the future. SWH has increased above 45°N in the North Atlantic (1.76 ±1.14 cm/year), corroborating what was concluded in the AR5 from ship observations and reanalysis-forced wave model hindcasts. In particular, a strong
positive anomaly of SWH and $P_{95}$ of SWH was found in this region in JFM 2022. However, contrary to Young & Ribal (2019), a strong decrease in SWH of nearly -2.1 ±0.76 cm/year has also been observed in the altimetric record over the last 19 years in JFM in the North Atlantic below 45°N. Moreover, all the trends of SWH and $P_{95}$ of SWH calculated in this study for JFM and JAS over 2002-2020 are much greater than those indicated by Young and Ribal (2019) over the period 1985-2018. The global maps of SWH extremes highlight the regions heavily affected by storms, such as the western North Pacific,
the North Atlantic and the tropical eastern Pacific. Trends in 100-year return levels seem to indicate an increase in wave levels linked to this energetic activity.

The L4 altimetric time series merges between one and four missions at a time. While the number of satellites doesn't impact the sign of the trends, it can affect their magnitudes, indicating that there is a need for a long, global and more homogeneous altimetric time series. Additionally, this study reveals the need for knowledge of uncertainties. A new product
was generated as part of this study to assess the effect of the number of satellites on the results. The conclusions given above nevertheless remain unchanged.

**Author contribution**

Alice Laloue conceptualized the statistical analysis and carried it out. Malek Ghantous provided the two-satellite altimetric

time series used in the discussion. Malek Ghantous, Yannice Faugère, Alice Dalphinet and Lotfi Aouf supervised the study

and provided guidance. Alice Laloue prepared the manuscript with contributions from Malek Ghantous.





**Competing interests**

The authors declare that they have no conflict of interest.

**Research funder**

Copernicus Marine Service

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
