# Peer review of "Statistical analysis of global ocean significant wave heights from satellite altimetry over the past two decades"

_State of the Planet, 2023_

## Author Comment (AC1)

[Figure]

**Figure 4 (revised): Effects of the number of satellites on the long-term trends in L4 altimetric time series. (a) Boxes in which regional trends were computed. Box 1: 30° N-43° N, 52° W-18° W; box 2: 66° S-47° S, 38° W-42° E; box 3: 60° S–35° S, 110° E-155° E. (b), (c), (d) Time series of daily mean SWH, of P95 daily maximum SWH, and of the daily number of observations in JFM averaged on a yearly basis, associated with each box. The bootstrap 95% confidence interval is represented with error bars. In red: the L4 multi-mission product (product reference 1), in black: L4 two-satellite product. Trends are represented by dashed lines when statistically significant for both products. Finally, the number of satellites combined in the multi-mission product is represented by coloured blocks as a function of time as in (Charles, 2021).**

[Figure]

**Supplement 1: 95th SWH percentile (a, b) climatology (2002-2020) and (c/d) annual trend (2002-2020) for both JFM (left column) and JAS (right column) from WAVERYS. Areas with anomaly above 1.5 times the interannual variability are outlined in black. Areas with trend statistically significant at the 95% level are outlined in black.**

[Figure]

**Supplement 2: SWH (a, b) climatology (2002-2020) and (c, d) annual trend (2002-2020), for both JFM (left column) and JAS (right column) from WAVERYS. Areas with anomaly above 1.5 times the interannual variability are outlined in black. Areas with trend statistically significant at the 95% level are outlined in black.**

---

## Author Response (AR1)

**Point-by-point response to the reviews**

**Relevant changes made in the manuscript were marked in red.**

**RC1**: 'Comment on sp-2023-25', Anonymous Referee #1, 07 Oct 2023

**Dear Reviewer,**

**We would like to thank you for your remarks and your comments, which have helped us to correct and improve the manuscript. Please find below our answers (marked in blue).**

General comments:  This study focuses on evaluating changes in significant wave height (swh) during the satellite altimeter period, making use of three different metrics:  (1) changes in mean swh, (2) changes in extrema, as measured by the 95$^{th}$ percentile, and (3) changes in the 100-year return amplitude.  All of these show increases over the study period (2002-2020).  This study is not the first to tackle this issue. One of the lingering concerns with altimeter-based studies is that changes in the number of satellites or the quality of satellite observations could result in false trend.  The authors have addressed this topic in their Figure 4 by comparing trends that would be obtained with two satellites vs the multi-satellite results.

Overall, I think this analysis is a useful contributor to the swh climate change discussion and it should be published after revision.

1.The results in Figure 4 are extremely useful but perhaps miss an opportunity to clarify exactly what mechanisms account for the differences between two-satellite and multi-satellite analyses. The text explains that the availability of more satellites increases the probability of seeing extreme events.  I think the point that isn't made clearly is that this difference occurs because swh has a non-Gaussian distribution, so that large extremes will alter the mean and 95$^{th}$ percentile statistics much more than extremely small events.  Although the authors show that this effect is not the defining driver of swh trends, the differences between the black and red lines in Figure 4 are important, and the authors should emphasize that the 2-satellite trends are potentially a more accurate measure of the long-term trends than the multi-satellite trends.

**We've added the clarifications you suggested to the relevant paragraphs.**

**Changes (clarifications):**

l.201-203: "For example, it is likely that more storms or extreme waves were sampled by the altimeters in the latter years of the period than in the former. The distribution of SWH is not Gaussian and is largely affected by extreme events, hence producing a spurious positive trend in SWH."

l.211-217: "The mean SWH is not greatly affected by the number of satellites and the trends of mean SWH are almost identical. On the other hand, the P$_{95}$ daily maximum SWH is sensitive to the increase in the number of observations and the multi-mission

product overestimates its trends compared with the two-satellite product. More importantly, the sign of the trend does not change, the spatial patterns of the trend are mostly consistent between the products and trends in the two-satellite product are contained within the uncertainty of trends in the multi-mission product. However, as the two-satellite product is more consistent over time, the long-term trends measured with it may be more accurate than those measured with the multi-mission product."

2.  There have been a number of studies on trends in significant wave height in recent years. This manuscript cites some altimeter-based analyses but also acknowledges some of the challenges in using a changing constellation of altimeters.  Another study by Bromirski et al (2023) that received some press took advantage of seismic data to show increasing waves and could corroborate these results.  The authors could consider whether to cite this:

Bromirski, P. D. (2023). Climate-induced decadal ocean wave height variability from microseisms: 1931–2021. *Journal of Geophysical Research: Oceans*, 128, e2023JC019722. https://doi.org/10.1029/2023JC019722

I've noticed recent work has evaluated seasonality of wave climatology and that could be pertinent for the current study.

**Thank you for the reference you sent us. Unfortunately, we decided not to include it because the comparison with the results presented could not provide a decisive conclusion.**

**The periods considered in this article are not the same as the one of the altimetric time series we use, and the regional patterns obtained in this article do not seem completely related to the patterns of SWH we obtained. The coarse spatial resolution of the altimetric time series may also take part in this conclusion.**

3.  The paragraph starting at line 143 appears to discuss Figure 3, but Figure 3 is not called out in the text. This should be clarified for readers.

**The correction has been made.**

4.  Line 147. The contrasting trends of SWH and P95 of SWH are not entirely clear. I think the sentence should perhaps say, "In the North Atlantic in JFM, both SWH and P95 of SWH show decreasing trends.  However, the 100-year return trends are largely positive."

**The correction has been made.**

5.  Lines 158-161. This paragraph discusses calculations using ERA5 and AVERYS data that appear not to be included in the manuscript. Either the paper should specify "(not shown)", of the figures should be added to supplementary information and called out in the text. Also true at line 218.

**The figures are not shown in the manuscript. The mention "(not shown)" was added.**

**You can still find the figures (fig1/2) associated with WAVERYS data in the supplementary information.**

6.  Line 171-177. I believe the figure references in this paragraph should be for Figure 4 rather than Figure 3. It's not clear to me why there are 5 boxes in Figure 4a, but only 3 highlighted in side panels, and I would suggest simplifying the figure to show only the 3 boxes that are included in time series analysis, or else add the additional panels in supplementary information and explain something about their significance.

**The study of regional trends was initially carried out on all the 5 boxes plotted in figure 4a. However, the regional trends calculated did not, in the end, provide any additional information to the 3 boxes selected in the side panels, so we decided not to display all of them.**

**Changes:**

Figure 4a was simplified as suggested to show only the 3 boxes that are included in the side panels.

7.  The discussion of Figure 4 indicates that the trends for the red and black lines agree within uncertainty estimates. I can see that this is true for the estimated trends, but I think the calculation would be more complete if the annual averages also included uncertainties. Does each individual year agree within uncertainties?

**Changes:**

Figures 4 (b) (c) (d): Uncertainties on the annual averages were added on the figures with error bars (Bootstrap 95% confidence intervals).

**On one hand, annual averages of daily mean SWH for both datasets (two-satellite in black and multi-mission in red) agree within uncertainties.**

**On the other hand, the annual average P95 of daily max SWH mostly agree within uncertainties. However, the bounds of the 95% confidence interval calculated for the average P95 of daily max SWH of the multi-mission product are systematically higher than the bounds of the confidence interval of the two-**

**satellite product. It is especially true from 2008, when the multi-mission product merges more missions at the same time, until the end of the period. The multi-mission product contains greater extreme SWH than the two-satellite product for those years.**

8.  References are not in alphabetical order and include some items that are not cited in the text. These should be cleaned up.

**The correction has been made.**

9.  Minor points of grammar/style

- line 8.  Remove "Indeed".  It doesn't make sense in this paragraph.
- lines 9-10.  Split the sentence into two.  "...several decades.  Such time series ..."  Otherwise "that" appears to reference "decades", which will confuse readers.
- line 20.  Change "such as" to "including"
- lines 21-22, 174-175.  Change "above" to "poleward of" or "north of".  Change "below" to "equatorward of" or "south of"
- lines 22-23.  "As for" is not a clear construction.  Maybe "The 100-year return levels of the SWH have significantly increased in the North..."
- line 27.  Add comma after "variability".
- Line 28.  Add comma after "changes"
- Line 33.  Split into two sentences, ending the first after "decades".  Then "Thus far these records have only been available in global ...."
- Line 34.  Remove "using"
- Lines 53-54.  "is the value above which 5% of the values in the mean time series fall".  This wording is not clear.  I would recommend rewriting, and in particular rewording "fall".
- Lines 58, 102.  Change to "Timmermans et al. (2020)"
- Line 73.  Split into two sentences:  "before.  Thus".  In English, this is called a "comma splice" and is not considered grammatically acceptable.
- Figure 2a labels.  "95e percentile" should be replaced with "95th percentile"
- Line 159.  Split into two sentences:  "datasets.  However,"
- Line 160.  It's unclear whether "they" refers to "ERA5 and WAVERYS" or something else.  The word "they" should be replaced with the appropriate noun.  I also suggest removing "As for extreme values", which is ambiguous.  I think the authors are using it to mean "Now we're changing

the topic to talk about extreme values", but it can be interpreted as "Similar to extreme values".

- Line 168.  Change to "The 100-year return levels have significantly increased in the North Atlantic and in the eastern North Pacific …."

- Line 186.  Add comma after "former".

- Lines 195-196.  Two consecutive sentences start with "However".  That makes too many contrasts, so at least one "However" should be removed.

- Line 197.  Change "doesn't" to "does not".  (No contractions in formal writing.)

- Line 209.  Start a new sentence with "However" to avoid a comma splice.

**Thank you for your careful proofreading. All minor points of grammar/style were corrected.**

**RC2**: ['Comment on sp-2023-25'](), Anonymous Referee #2, 24 Jan 2024

**Dear Reviewer,**

**We would like to thank you for your remarks and comments, which have helped us to clarify and improve the manuscript. Please find below our answers (marked in bold).**

This paper presents an investigation of the trends to the significant wave height based on the combined Copernicus L4 altimeter product comprising seven altimeter missions.

Abstracts should be in the present tense, avoid the use of abbreviations and ideally should not have references. It should succinctly summarise the findings in the paper. Please remove motivational text like "The analysis of global ocean surface waves and of long-term changes is important to climate research".

**The abstract was rewritten in the present tense; most abbreviations were suppressed, and the reference was suppressed as you suggested. We removed the initial motivational text. Please find below the revised abstract.**

**Changes (abstract):**

**Abstract was corrected as suggested.**

"The analysis of global ocean surface waves and of long-term changes requires accurate time series of waves over several decades. Such time series have previously only been available from model reanalyses or from in situ observations. Now, altimetry provides a long series of observations of significant wave heights (SWHs) in the global ocean. The aim of this study is to analyse the climatology of significant wave heights and extreme significant wave heights derived from remote sensing in the global ocean and their long-term trends from 2002 to 2020 using different statistical approaches as the mean, the 95th percentile and the 100-year return level of SWH. The mean SWH and the 95th percentile of SWH are calculated for two seasons: January, February and March, and July, August and September and for each year. A trend is then estimated using linear regression for each cell in the overall grid. The 100-year return levels are determined by fitting a Generalised Pareto distribution to all exceedances over a high threshold. The trend in 100-year return level is estimated using the transformed-stationary approach, which, to our knowledge, is used for the first time to draw a global map based on altimetry. Predominantly large positive

trends over 2002-2020 for both SWH and extreme SWH are mostly found in the southern hemisphere, including the South Atlantic, the Southern Ocean and the southern Indian Ocean, which is consistent with previous studies. In the North Atlantic, SWH has increased poleward of 45°N, corroborating what was concluded in the fifth IPCC Assessment Report, however SWH has also largely decreased equatorward of 45°N in wintertime. The 100-year return levels of SWH have significantly increased in the North Atlantic and in the eastern tropical Pacific, where the cyclone tracks are located. Finally, in this study we find trends of SWH and 95$^{th}$ percentile of SWH over 2002-2020 to be much higher than those indicated in the literature for the period 1985-2018."

The paper is quite well written and is a nice summary of the Copernicus L4 product. I think it is important to acknowledge that although the series represents a high-quality data set of altimeter measurements, it is still very short for EVA. The data set considered by Ribal and Young (2019) was much longer, covering the period 1985-2018. Would it be possible to combine the two?

**As you noted, a series of 19 years, even of high quality, is still very short for EVA. This is why we also considered merging the two series from the start of our study and we contacted the two authors. Unfortunately, it may not be possible to combine the two series as they are calculated differently.**

**In the discussion we emphasize the differences that appear between the time series resulting from the multi-mission product (which is not consistent over time, merging from 1 to 4 satellites at a time) and the time-series from a product combining only two satellites. The differences appear even more when we consider the extreme values of SWH.**

**Combining the two time series could add even more heterogeneities in the time series and impact the resulting values. This is why we restricted ourselves to the 19-year-old series.**

The claim that "The EVA allowed us to study 100-year SWH with only a 19-year long altimetric time series. All the values of SWH exceeding the 95th percentile and separated by at least 72h were selected according to the peaks-over-threshold method" grates on me. It is not so that the transformation to a trend-free series in itself will get you off the hook. The period 2002-2018 could still be exceptional compared to a much longer series if slowly changing processes are at play, or by pure coincidence. I would like to see 95% confidence estimates or credibility intervals (if a Bayesian approach is taken). Please include a more detailed description of the transformed stationary method as I'm sure the casual reader will not be too disturbed by a couple of equations. More importantly, I would like to see a discussion of the weaknesses of this method, and in particular, re my previous comment, what are your concerns when applying it to such a short series?

A 95% confidence interval was estimated for the 100-year return level of SWH (displayed Fig 3a). The differences between the 100-year return level and the corresponding lower bound and upper bound were displayed separately Fig 3.c and Fig 3.d. The largest confidence intervals are found in the typhoon region where the greater return levels were found.

**Changes:**
Figure 3: Figure 3 (c) and Figure 3 (d) were added.

**We included a more detailed description of the transformed stationary method.**

**Please find below the revised paragraph.**

**Changes:**
l. 68-86: Paragraph was clarified, and several equations and explanations were added.

"All the values of SWH exceeding the 95$^{th}$ percentile and separated by at least 72h were selected according to the peaks-over-threshold method. A Generalised Pareto Distribution (GPD) could then be fitted to the exceedances (see equation below). The return levels associated with the 100-year return period were estimated from this GPD.

$$F(x) = 1 - [1 + \frac{\xi(x - \mu)}{\sigma}]^{-\frac{1}{\xi}}$$

With $\mu$ , $\xi$ and $\sigma$ are the location, shape and scale parameters.

The EVA has a major disadvantage in that it usually requires the time series to be stationary. The transformed-stationary approach overcomes this issue by transforming the non-stationary altimetric time series $y(t)$ into a stationary one $x(t)$ through standardization (Eq.1). The EVA is then applied to $x(t)$, and the location $\mu_x$ and scale $\sigma_x$ parameters of the GPD are estimated by maximizing the likelihood function. The reverse transformation (Eq. 2, 3) is finally used to recover the time-varying parameters $\mu_y(t)$ and $\sigma_y(t)$ associated with $y(t)$, enabling us to obtain the non-stationary extreme SWH distribution and to assess its trend. The transformation from y(t) to x(t) and the reverse transformation of the shape, location and scale parameters associated with the non-stationary series are given by:

$$x(t) = \frac{y(t) - T_y(t)}{S_y(t)} \tag{1}$$

$$\mu_y(t) = S_y(t)\mu_x + T_y(t) \tag{2}$$

$$\sigma_y(t) = S_y(t)\sigma_x \tag{3}$$

$$\xi_y = \xi_x \tag{4}$$

where $T_y(t)$ and $S_y(t)$ are the trend and the standard deviation of $y(t)$, and $\mu_x$, $\xi_x$ and $\sigma_x$ are the parameters associated with the stationary series which are not dependent on time."

**As you suggested we added a small discussion addressing your concerns, in particular on the issue of applying EVA to such a short series as you explained. Find below that corresponding discussion.**

**Changes:**
l.234-239: A paragraph was added in the discussion.
  "Finally, the EVA gave us a good initial estimate of SWH extremes based on altimetry measurements, in line with the literature. However, these results must be treated with caution, as the altimeter series is very short (less than twenty years), so few measurements could be selected to estimate the GPD parameters. Similarly, the measurement period is not necessarily representative of a longer time series. This ultimately leads to large confidence intervals for the extreme values estimated. In addition, the transformed-stationary approach used assumes that the GPD shape parameter is constant, which is valid in most cases but may prove false in some."

I think the paper may be acceptable given a major revision which addresses these concerns.

Specific (minor) comments:
 **Thank you for your careful proofreading.**
Abstract: The first line, "The analysis of global ocean and coastal applications. Indeed, waves contribute to flooding, coastal erosion, extreme sea level events and ocean circulation. They also play a role in air-sea and sea-ice interactions." This is better suited for the introduction.

**The first line of the abstract was suppressed as it was redundant with the introduction.**

**Changes:**

l.1: First sentence suppressed.

Abstract: "Specify in the abstract that significant wave height is meant when stating "climatology of wave heights and extreme wave heights"

**The correction has been made.**

L 73: hasn't -> has not

**The correction has been made.**

L 75: (Herbasch et al., 2023) should be Hersbach et al (2020). This error is repeated elsewhere.
**The correction has been made.**

L 158: "For comparison, the same figures were produced using ERA5 and WAVERYS data." Where?

**We decided not to show the figures produced using ERA5 and WAVERYS as they were in line with the existing literature and did not add any new information. The mention "(not shown)" was added after this sentence.**

**However, you may find complementary figures displayed for WAVERYS in supplements.**

**You shall find later the revised manuscript.**